# Can We Further Elicit Reasoning in LLMs? Critic-Guided Planning with Retrieval-Augmentation for Solving Challenging Tasks

## Abstract

State-of-the-art large language models (LLMs) exhibit impressive problem-solving capabilities but may struggle with complex reasoning and factual correctness. Existing methods harness the strengths of chain-of-thought (CoT) and retrieval-augmented generation (RAG) to decompose a complex problem into simpler steps and apply retrieval to improve factual correctness. These methods work well on straightforward reasoning tasks but often falter on challenging tasks such as competitive programming and mathematics, due to frequent reasoning errors and irrelevant knowledge retrieval. To address this, we introduce **C**ritic-guided planning with **R**etrieval-augmentation, CR-Planner, a novel framework that leverages fine-tuned critic models to guide both reasoning and retrieval processes through planning. CR-Planner solves a problem by iteratively selecting and executing sub-goals. Initially, it identifies the most promising sub-goal from reasoning, query generation, and retrieval, guided by rewards given by a critic model named sub-goal critic. It then executes this sub-goal through sampling and selecting the optimal output based on evaluations from another critic model named execution critic. This iterative process, informed by retrieved information and critic models, enables CR-Planner to effectively navigate the solution space towards the final answer. We employ Monte Carlo Tree Search (MCTS) to collect the data for training the critic models, allowing for a systematic exploration of action sequences and their long-term impacts. We validate CR-Planner on challenging domain-knowledge-intensive and reasoning-heavy tasks, including competitive programming, theorem-driven math reasoning, and complex domain retrieval problems. Our experiments demonstrate that CR-Planner significantly outperforms baselines, highlighting its effectiveness in addressing challenging problems by improving both reasoning and retrieval. [1]

## 1 Introduction

State-of-the-art large language models (LLMs), while demonstrating remarkable problem-solving capabilities (OpenAI, 2023; Cheng et al., 2023), still face two key challenges: reasoning for complex tasks (Huang et al., 2024) and domain-specific knowledge (Zhao et al., 2023a). Existing approaches (Yao et al., 2023b; Zhao et al., 2023b; Li et al., 2024) seek to harness the strengths of both chain-of-thought (CoT) reasoning (Wei et al., 2022) and retrieval-augmented generation (RAG) (Lewis et al., 2020) on knowledge-intensive complex reasoning problems. Specifically, instead of invoking RAG solely at the initial stage, these methods can potentially apply RAG at each reasoning step. This integrated approach enhances both retrieval and reasoning, as the insights gained from reasoning enable the retrieval of more relevant information, while the retrieved knowledge improves the factuality of the subsequent reasoning steps of the model. To better incorporate retrieval into reasoning, some methods, such as Self-RAG (Asai et al., 2024) and its variants (Yan et al., 2024; Islam et al., 2024), directly finetune LLMs to decide when to retrieve and whether to adopt the retrieved documents by adding special reflection tokens.

---

[1]We will make our code and data publicly available.

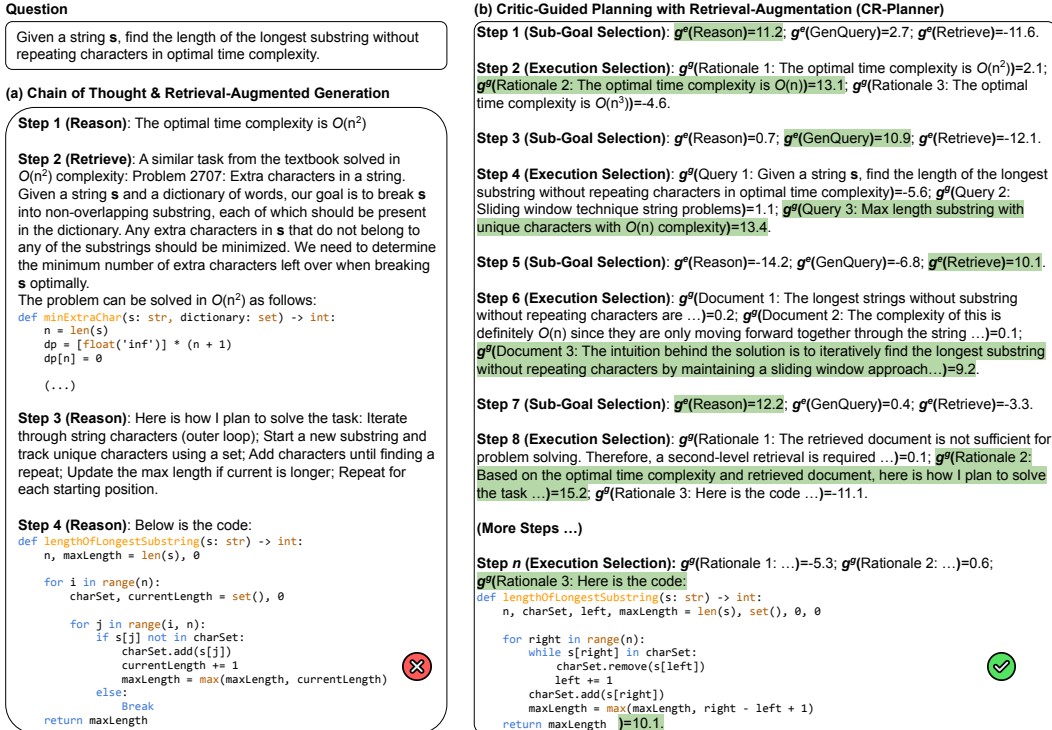

Figure 1: Comparison between (a) chain-of-thought reasoning (Wei et al., 2022) with retrieval-augmented generation (Lewis et al., 2020) and (b) critic-guided planning with retrieval-augmentation or CR-Planner (this work). $g(\cdot)$ indicates the critic model (or value function) that assigns a reward (or value) to an action (see Equation 2). Texts in (b) highlighted in green are actions selected at each step. For succinct presentation, only pivotal steps are shown in the figure.

While the above methods have shown prospects, they are generally limited to handling problems with relatively simple reasoning processes, such as answering two-hop questions like "What year was the Argentine actor who directed El Tio Disparate born?" These methods often fail to solve **domain-knowledge-intensive** and **reasoning-heavy** problems, such as competitive programming problems (Shi et al., 2024) which require the model to possess rich algorithmic knowledge and strong reasoning capability. Specifically, these methods often struggle with two significant types of errors, as shown in Figure 1 (a). The first is **reasoning error**. When presented with the problem "Given a string **s**, find the length of the longest substring without repeating characters in optimal time complexity," a CoT approach may incorrectly generate that "The optimal time complexity is $O(n^2)$" in its initial reasoning step. This erroneous reasoning step then cascades through subsequent steps, leading to an incorrect final answer. The second type of error is **retrieving error**. The effectiveness of the retrieval process depends on the accuracy of the generated search queries and the selection of the retrieved documents. If the preceding reasoning step is flawed, the query generator could be misguided, leading the retriever to return misinformation, as shown in Figure 1 (a). Additionally, the selection of retrieved documents could be erroneous. Thus, the subsequent reasoning will be grounded on a wrong prior.

To address these errors, we present critic-guided planning with retrieval-augmentation (CR-Planner), a framework designed to tackle reasoning-heavy problems requiring extensive domain knowledge. CR-Planner systematically plans both reasoning and retrieval processes with specially fine-tuned critic models. An example of CR-Planner in action is illustrated in Figure 1 (b), using the question mentioned above. CR-Planner begins with **Sub-Goal Selection**, where it selects a sub-goal from three options: REASON (generating rationales), GENQUERY (generating search queries), and RE-TRIEVE (retrieving documents), based on reward scores estimated by a critic model, *the sub-goal critic*. After choosing the sub-goal of REASON in Step 1, CR-Planner proceeds to **Execution Selection**, where it samples several candidate rationales for the next step. Another critic model, *the*

*execution critic* is then employed to select the optimal rationale, which in this case is "The optimal time complexity is $O(n)$." In Step 3, CR-Planner returns to sub-goal selection to determine the next best sub-goal. This iterative process of alternating between sub-goal selection and execution selection continues until the final answer is reached, with each step effectively guided by the corresponding critic model. Regarding the implementation, CR-Planner incorporates two types of LLMs: a large general generator model (*e.g.,* GPT-4) and small critic models (*e.g.,* Llama-3-8B) fine-tuned with domain-specific *(critiquing)* knowledge. Specifically, when executing a sub-goal, the generator model generates multiple candidate executions (*e.g.,* rationales or search queries, depending on the current sub-goal type). Then, an execution critic corresponding to the sub-goal type performs planning by selecting the most prospective option. Such a design allows CR-Planner to leverage the generation and reasoning strengths of large generalist LLMs and meanwhile, its small critic models are easier to train with domain-specific *(critiquing)* knowledge.

To optimize the planning performance for sub-goal and execution selection in each domain, we train the critic models separately. The training process for these critic models requires the collection of reasoning and retrieval trajectories with step-wise reward labeling. However, the availability of such data is limited, and annotating it with humans poses significant costs (Lightman et al., 2024). To address this data scarcity, we utilize Monte Carlo Tree Search (MCTS) (Browne et al., 2012) for efficient data collection. MCTS estimates long-term expected rewards at each step by comparing simulated outcomes with gold labels and propagates the rewards back to the previous steps. By simulating multiple possible trajectories, we can get reliable rewards for each step, thereby effectively training the critic models to guide the reasoning and retrieval process at each step.

In summary, our key contributions are: (1) We introduce CR-Planner, a novel framework designed to tackle domain-knowledge-intensive and reasoning-heavy problems by employing specially fine-tuned critic models that guide both reasoning and retrieval processes through planning; (2) We propose using MCTS to effectively collect training data for the critic models, enhancing their ability to estimate the long-term impact of an action. (3) We perform extensive experiments on challenging tasks that require domain knowledge and complex reasoning, including competitive programming, math reasoning, and complex retrieval. CR-Planner outperforms the baseline by 10.06% on average.

## 2 CRITIC-GUIDED PLANNING WITH RETRIEVAL-AUGMENTATION

We introduce the critic-guided planning with retrieval-augmentation framework (CR-Planner) to address challenging tasks that are both domain-knowledge-intensive and reasoning-heavy. As shown in Figure 2, CR-Planner operates with two key components during inference: **(1) Sub-Goal Selection**: Given the current state, it employs a sub-goal critic model to determine the sub-goal among REASON, GENQUERY, and RETRIEVE that leads towards the desired answer. **(2) Execution Selection**: Upon selecting a sub-goal, CR-Planner undertakes multiple possible executions to realize the sub-goal. For instance, it may generate multiple search queries to achieve the GENQUERY sub-goal. Then, an execution critic model specifically designed to assess the executions for the sub-goal is employed to select the optimal execution among these candidates. In the above process, a general generator model collaborates with multiple specialized critic models to address the task effectively. We leverage the strengths of the generator model to generate initial plans, while the specialized critic models are fine-tuned to guide optimal routing. To ensure that the training data for the critic models is comprehensive and represents global reward information, we employ Monte Carlo Tree Search (MCTS) to collect the training data.

### 2.1 PROBLEM FORMULATION

We formally define the associated planning environment of CR-Planner as a Markov Decision Process (MDP) represented by the tuple $(\mathcal{S}, \mathcal{A}_s, \mathcal{P}, \mathcal{R}, T)$, where:

- $\mathcal{S}$ represents the state space. Specifically, the state at timestamp $t$, denoted by the random variable $s_t$, comprises a action-observation trajectory history $(o_0, a_0, ..., a_{t-1}, o_t)$, where $a_{t-1}$ is the action taken at timestep $t-1$, and $o_t$ is the observation made after that. The observation $o$ can be REASON, GENQUERY, or RETRIEVE in sub-goal selection stage, and RATIONALE, QUERY, or DOC in execution selection stage. Additionally, a state is named after its last observation, *e.g.,* RATIONALE state $s_t$ means $o_t$ is a RATIONALE.

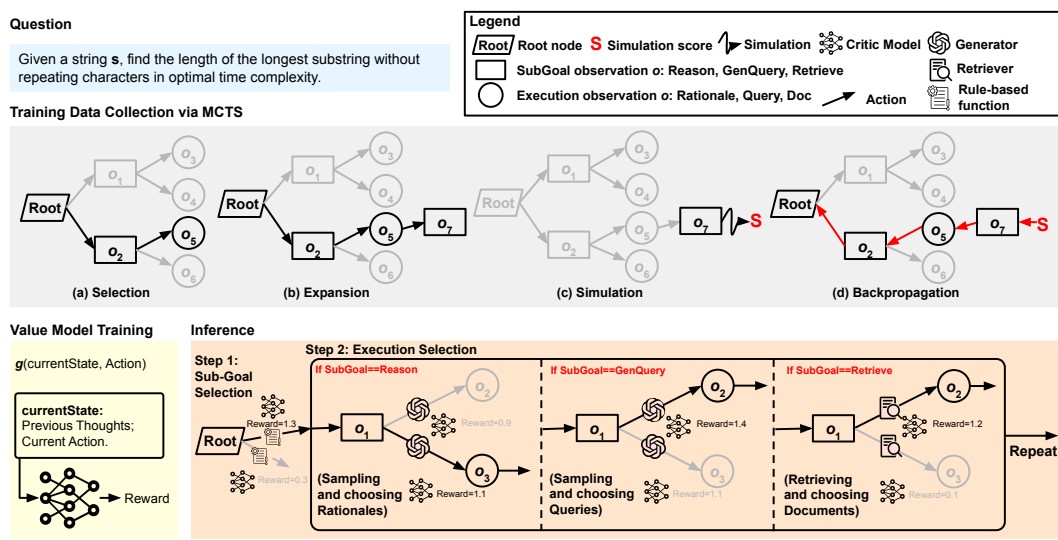

Figure 2: The retrieval-augmented and critic-guided planning (CR-Planner) framework. The figure illustrates training data collection via MCTS, critic model training, and inference. For succinct presentation, SUBGOAL observations (REASON, GENQUERY, and RETRIEVE) are shown as labeled rectangles and EXECUTION observations (RATIONALE, QUERY, and DOC) as labeled circles. A state $s_t$ includes all preceding nodes (observations) and arrows (actions) up to the last node.

- $\mathcal{A}_s$ represents the actions available at each state. For example, the actions available at the sub-goal selection stage, *i.e.,* at the Root state or after observing an outcome of an execution selection are: *reasoning*, *querying*, and *retrieving*. The possible actions available at the execution selection stage arise from the sampling for the corresponding sub-goal (*i.e.,* temperature sampling for REASON and GENQUERY, and top-k candidates for RETRIEVE). For example, Steps 1 and 2 in Figure 1 (b) illustrate the REASON and RATIONALE observations generated following the sub-goal selection and execution selection stages, respectively.

- The state transition $\mathcal{P}$ defines how the states evolve after an action is taken. In our context, state transitions are determined and handled by different functions depending on the current state. During the execution selection stage, a REASON or GENQUERY state transits to the respective RATIONALE or QUERY execution outcomes via the distribution defined by a large general generator model $f_{gen}(\cdot)$. Similarly, a RETRIEVE state transits to a DOC state via a retriever $f_{retr}(\cdot)$. During the sub-goal selection stage, the transition is more straightforward and done via a rule-based function $f_{rule}(\cdot)$, *e.g.,* selecting *reasoning* action transits to a REASON state.

- The reward function $\mathcal{R}(s_t, a)$ specifies the expected reward received after taking an action $a_t$ at state $s_t$. In our context, fine-tuned critic models estimate the rewards and guide the decision-making process by encouraging actions that contribute the most towards solving the MDP. Details of the critic models are provided in Section 2.2.

- Lastly, $T$ represents the maximum number of steps that can occur within the MDP.

Solving the MDP requires generating an optimal plan in the form of a trajectory: $\tau* = (s_0, a_0, ..., s_t, a_t, ..., s_{T-1}, a_{T-1}, s_T)$ that maximizes the total expected rewards. [2]

## 2.2 INFERENCE OF CR-PLANNER

When tackling domain-knowledge-intensive and reasoning-heavy problems, errors may occur during the reasoning process, which can then propagate to subsequent steps. Therefore, ensuring accuracy at each step of the process from the very beginning is essential. Additionally, external information is not always necessary in the problem-solving process. In fact, deciding when to access

---

[2]Details of state types and action spaces are in Appendix C Table 6.

external sources is a critical decision (Asai et al., 2024). Furthermore, as highlighted by Li et al. (2024), a significant challenge in RAG is the accuracy of the retrieval process itself. Consequently, it is crucial to ensure both the quality of search queries and the selection of retrieved documents. To address these challenges, our method employs critic models at each time step to guide the decision-making process. Specifically, at time step $t$, given the current state $s_t$, the critic model $g$ assesses the available actions $\mathcal{A}_{s_t}$ and helps select an action $a_t$ that maximizes the expected reward.

**Action selection using the critic models.** At timestamp $t$, the policy model $\pi$ determines the next action as:

$$a_t = \pi(s_t) = \arg \max_{a \in \mathcal{A}_{s_t}} \mathcal{R}(s_t, a). \tag{1}$$

The action space $\mathcal{A}_{s_t}$ varies depending on $s_t$. As previously discussed in Section 2.1 and outlined in Table 6, for a state in the sub-goal stage, the action space leads to possible executions of that sub-goal, while for a state in the execution stage, the action space leads to the possible subsequent sub-goals. $\mathcal{R}(s_t, a)$ is the expected reward when taking action $a$ in state $s_t$ and estimated by the critic models:

$$\mathcal{R}(s_t, a) = \begin{cases} g^e_{\text{RATIONALE}}(s_t, a), & \text{if } s_t = \text{REASON state} \\ g^e_{\text{QUERY}}(s_t, a), & \text{if } s_t = \text{GENQUERY state} \\ g^e_{\text{DOC}}(s_t, a), & \text{if } s_t = \text{RETRIEVE state} \\ g^g(s_t, a), & \text{otherwise.} \end{cases} \tag{2}$$

Specifically, distinct critic models are utilized for different state types: $g^g(\cdot)$ is for determining the next sub-goal at the current execution state (*i.e.,* the inference Steps 1 in Figure 2), and $g^e(\cdot)$ is for evaluating different execution candidates at the current sub-goal state (*i.e.,* the inference Step 2 in Figure 2). Additionally, according to the sub-goal states, $g^e(\cdot)$ has three variants $g^e_{\text{RATIONALE}}, g^e_{\text{QUERY}},$ and $g^e_{\text{DOC}}$, correspondingly evaluating rationales, queries and the retrieved documents.

**State transition with the selected action.** Once $a_t$ is determined and executed, the state is then transited from $s_t$ to $s_{t+1} = (s_t, a_t, o_{t+1})$, where

$$o_{t+1} = \begin{cases} f_{gen}(s_t, a_t), & \text{if } s_t = \text{REASON or GENQUERY state} \\ f_{retr}(s_t, a_t), & \text{if } s_t = \text{RETRIEVE state} \\ f_{rule}(s_t, a_t), & \text{otherwise.} \end{cases} \tag{3}$$

As mentioned in Section 2.1, given the current state $s_t$ and action $a_t$, we employ three specific functions to generate different types of outcomes. The generator $f_{gen}(\cdot)$ generates either a RATIONALE or QUERY. The retriever $f_{retr}(\cdot)$ outputs a DOC. Last but not least, the rule-based function $f_{rule}(\cdot)$ outputs a SUBGOAL. The SUBGOAL is a predefined natural language. For example, a REASON thought is "The next step is to generate a rationale".

**Termination conditions and the final answer.** This process continues until one of two conditions is met. The process ends at step $t$ if the observation $o_t$ includes the complete answer. Otherwise, if $t$ equals $T$ and $o_t$ does not contain the final answer, an extra step occurs to force the model to conclude the answer. In this case, a concluding answer is generated using an LLM.

## 2.3 THE CRITIC MODELS

The CR-Planner framework relies heavily on its critic models as key components. These models evaluate actions and steer the overall process of sub-goal and execution selection. To fulfill this role effectively, the critic models must accurately assess each action based on its potential contribution to the entire problem-solving process. As such, the collection of training data for the critic models is crucial, for which we utilize Monte Carlo Tree Search (MCTS). MCTS is particularly well-suited for generating training data for the critic models due to its ability to explore the long-term impacts of potential actions while balancing exploration and exploitation. By simulating numerous possible action-observation trajectories, MCTS can provide a rich dataset of both successful and unsuccessful trajectories, helping the critic model learn to differentiate effective actions from suboptimal ones.

**Collecting data via MCTS.** As shown in Figure 2, MCTS consists of the four key steps: **(1) Selection.** Starting from the `root` state $s_0$, the algorithm selects child node (with observation) recursively based on the Upper Confidence Bound (UCB1) that balances exploration and exploitation. The UCB1 value for $o_i$ is computed as $\frac{v_i}{n_i} + c\sqrt{\frac{\ln n_p}{n_i}}$, where $v_i$ is the cumulative rewards of $o_i$, $n_i$ is the number of times $o_i$ has been visited, and $n_p$ denotes the number of visits to the parent thought of $o_i$. This process continues until it reaches a node that is not fully expanded or a terminal node. **(2) Expansion.** If the selected $o_i$ is not terminal and has unexplored child nodes, MCTS expands the tree by adding one or more of these unexplored child nodes. This represents exploring new actions available from the current action space $\mathcal{A}_{s_t}$. **(3) Simulation.** From the newly added observation, MCTS simulates a playthrough to a terminal state by employing a generative model $f_{gen}(\cdot)$ to generate the final answer based on existing observations. This simulation estimates the potential outcome from the observation. **(4) Backpropagation.** The result of the simulation is then propagated back up the tree. Each node along the path to the root updates its statistics, including visit counts and total reward, which informs future selection decisions by reflecting the observed outcomes. For each data point in the training dataset, we run MCTS for $N$ steps and collect pairwise data from the final state for each observation type. In particular, a chosen observation $o_i$ is the one with the highest score, while a rejected observation $o_i'$ is one of the observations sharing the same parent node but a lower score. For critic model $g_{\text{RATIONALE}}^e(\cdot)$, we collect $\mathcal{D}^{\text{RATIONALE}} = \{(O_i^{\text{RATIONALE}}, o_i, o_i')...\}$, where $O_i^{\text{RATIONALE}}$ represents previous RATIONALEs along the trajectory before the current RATIONALE $o_i$. It is crucial to evaluate $o_i$ considering all prior rationales. The critic model $g_{\text{QUERY}}^e(\cdot)$ uses $\mathcal{D}^{\text{QUERY}} = \{(o_i^{\text{RATIONALE}}, o_i, o_i')...\}$, where $o_i^{\text{RATIONALE}}$ is one immediately preceding RATIONALE of QUERY $o_i$. For the critic model $g_{\text{DOC}}^e(\cdot)$, we have $\mathcal{D}_i^{\text{DOC}} = \{(o_i^{\text{RATIONALE}}, o_i^{\text{QUERY}}, o_i, o_i')...\}$, where $o_i^{\text{RATIONALE}}$ and $o_i^{\text{QUERY}}$ are the immediately preceding RATIONALE and QUERY of DOC $o_i$. Lastly, the SUBGOAL critic model $g^g(\cdot)$ uses $\mathcal{D}^{\text{SUBGOAL}} = \{(O_i, o_i, o_i')...\}$, where $O_i$ represents all previous observations of any type along the trajectory.

**Training.** For each of the collected training datasets described above, we train a dedicated critic model as shown in Figure 2. Following Burges et al. (2005) and Ouyang et al. (2022), we employ pairwise ranking loss to optimize the parameters.

## 3 EXPERIMENTS

### 3.1 SETUP

**Models.** In our experiments, we employ GPT-4 (`gpt-4o-2024-05-13`) as the black-box LLM for generation during both inference and training data collection. Since CR-Planner requires the sampling of diverse RATIONALE and QUERY, we set the decoding temperature to 0.7. To ensure training and inference efficiency, we limit the sampling to three instances due to cost concerns. For the critic models, we fine-tune `Skywork-Reward-Llama-3.1-8B` (Skywork, 2024) with LoRA (Hu et al., 2021), which was trained as a sequence classifier with the Skywork Reward Data Collection and excels at scoring in complex scenarios, such as mathematics and coding. The first logit value of the model output is used as the reward score of our critic models.

**Baselines.** We compare CR-Planner with both commonly used baselines and state-of-the-art methods to offer a comprehensive evaluation: **(1) Standard prompting (Standard)** (Ouyang et al., 2022), which directly generates the answer. **(2) Chain-of-Thought (CoT)** (Wei et al., 2022), which generates multiple rationales before the final answer to enhance the models' reasoning ability. **(3) Reflexion** (Shinn et al., 2023), a framework uses linguistic feedback to further improve models' reasoning. **(4) Standard retrieval-augmented generation (RAG)** (Lewis et al., 2020), which retrieves relevant knowledge based on the problem itself and then lets the model to generate the final answer using both the problem and the retrieved knowledge. **(5) Chain-of-Knowledge (CoK)** (Li et al., 2024), a state-of-the-art CoT-based framework designed to enhance prediction accuracy by retrieving and post-editing rationale at each step. [3] All methods are zero-shot by default unless otherwise specified.

---

[3]We exclude Self-RAG as a baseline because it requires training the base model, which is not feasible in our setup. This further highlights the flexibility of CR-Planner.

## 3.2 COMPETITIVE PROGRAMMING

**USACO benchmark.** Computing Olympiads require complex algorithmic reasoning, puzzle-solving skills, and the ability to generate efficient code. Furthermore, retrieving knowledge from programming textbooks and similar problems from a problem bank can aid in solving these problems. In this sub-section, we employ the USACO benchmark (Shi et al., 2024), which includes 307 problems from the USA Computing Olympiad, to evaluate the performance of CR-Planner and baseline methods in the domain of competitive programming. USACO problems are categorized into four difficulty levels (*i.e.,* 123 bronze, 100 silver, 63 gold, and 21 platinum problems) and test various core skills, including complete search, binary search, and segment tree implementation. Typically, solving a USACO problem involves several steps: restating the problem in simple terms since many are framed within real-world contexts; retrieving relevant knowledge from textbooks or similar problems from a problem bank; conceptualizing the solution in plain English; drafting a pseudocode solution; and finally, producing the complete Python solution with comments. Thus, the USACO benchmark is an ideal fit for evaluating both the complex reasoning and retrieval capabilities of the models, making it highly relevant to this paper.

**External knowledge.** Following the baseline methods outlined in the USACO benchmark (Shi et al., 2024), we use both textbooks and a problem bank as external knowledge sources. The textbooks consist of 30 human-written chapters covering algorithmic concepts tailored specifically for the USA Computing Olympiad. The problem bank includes all other USACO problems except for the one currently being solved. Following Shi et al. (2024), we employ both textbooks and the problem bank as external sources for all methods. Additionally, we employ a BM25 retriever to execute the retrieval process, obtaining relevant information from external knowledge sources.

**Results and observations. (1) CR-Planner outperforms all baselines consistently.** Table 1 presents the results for USACO using various methods. CR-Planner significantly outperforms all baseline methods, achieving a 7.49% improvement in overall performance compared to standard prompting. This highlights the effectiveness of CR-Planner. **(2) Reasoning-driven methods offer limited improvements.** We observe that reasoning-driven methods like CoT and Reflexion do improve the performances of

Table 1: Pass@1 performances on USACO. The *Retrieval+Reflection\** result is from Shi et al. (2024).

| Method | Bronze | Silver | Gold | Platinum | Overall |
|---|---|---|---|---|---|
| Standard | 18.70 | 6.00 | 3.17 | 0.00 | 10.10 |
| CoT | 21.95 | 8.00 | 4.76 | 0.00 | 12.38 |
| RAG | 17.07 | 4.00 | 1.59 | 0.00 | 8.47 |
| CoK | 15.45 | 5.00 | 1.59 | 0.00 | 8.14 |
| CR-Planner | **26.02** | **10.00** | **14.29** | **14.29** | **17.59** |
| Reflexion | 23.58 | 9.00 | 4.76 | 0.00 | 13.36 |
| *Retrieval+Reflexion\** | - | - | - | - | *18.05* |
| CR-Planner+Reflexion | **34.15** | **16.00** | **14.29** | **14.29** | **22.80** |

the standard prompting method on bronze, silver, and gold problems, reaffirming that intermediate rationales and critique-based reasoning aid in solving reasoning tasks (Wei et al., 2022; Shinn et al., 2023). However, the improvements are trivial, and these methods fail to improve performance on platinum-level problems. We attribute this to the model's limited knowledge of the tasks or the generation of faulty rationales and critiques. **(3) Faulty retrieval hinders performance.** We observe that both standard RAG and CoK perform worse than the standard prompting method, consistent with the findings of Yao et al. (2023b) and Shi et al. (2024). This decline in performance can be attributed to the quality of retrieval. As demonstrated in Figure 1, if the retrieved example is irrelevant to the original problem, it may mislead the model into generating an incorrect answer. Additionally, we notice that CoK performs worse than RAG due to its reliance on multiple retrievals at individual steps, increasing the likelihood of misleading information being introduced and leading to a faulty final answer. **(4) CR-Planner improves harder problems.** CR-Planner notably boosts the performances on gold- and platinum-level problems. As aforementioned, while CoT offers minor improvements, it falls short on more difficult problems, and retrieval can hinder performance due to irrelevant knowledge. In contrast, CR-Planner employs critic models to guide both the reasoning and retrieval through the process, leading to non-trivial improvements at the two highest levels of programming problems. **(5) CR-Planner is orthogonal with other methods.** Reflexion executes the initially generated code and uses the execution results of a few test cases as linguistic feedback to revise the code. CR-Planner works orthogonal with such methods, leading to a significant improvement of 9.44%, further highlighting the effectiveness of critic-guided planning with retrieval-augmentation.

### 3.3 THEOREM-DRIVEN MATH PROBLEMS

**TheoremQA-Maths.**  When tackling a new theorem-driven math problem, people often reference solved problems with similar reasoning logic. However, finding such problems can be challenging because even if two problems share similar reasoning logic, they might appear very different on the surface. Moreover, in theorem-driven math problems, the reasoning process is critical. A single flawed step in the logic can lead to wrong subsequent rationales and finally an incorrect final answer. In this sub-section, we use the rewritten Math set from TheoremQA (Chen et al., 2023), named TheoremQA-Math, as introduced in the BRIGHT dataset (Su et al., 2024). TheoremQA-Math consists of 206 solvable questions that have been improved for fluency and coherence, with all questions requiring the application of math theorems (*e.g.,* the binomial theorems). To solve a problem in the TheoremQA-Math dataset, the process typically involves the following steps: understanding and restating the problem in simple terms; retrieving relevant knowledge from solved problems; conceptualizing the solution in plain English; and finally, generating the solution. Solving problems from TheoremQA-Math requires both complex reasoning and knowledge of Math theorems, making it pertinent to this paper.

**External knowledge.**  Following the BRIGHT benchmark (Su et al., 2024), we employ a collection of processed documents sourced from high-quality STEM datasets, including GSM8K (Cobbe et al., 2021), GSM8K-RFT (Yuan et al., 2023), MATH (Hendrycks et al., 2021), AQuA-RAT (Ling et al., 2017), TheoremQA (Chen et al., 2023) and CAMEL-MATH (Li et al., 2023). To ensure efficient retrieval during both the training data collection and inference stages, we opt for the term-based retrieval method BM25, similar to what is used in competitive programming.

**Results and observation.**  Similar to competitive programming, as shown in Table 2, we observe a notable performance improvement from CR-Planner, with 13.59% on TheoremQA-Math compared to standard prompting method. This further demonstrates the effectiveness of CR-Planner in tasks requiring knowledge retrieval and complex reasoning. Interestingly, Reflexion exhibits inferior performance compared to CoT, which we attribute to Reflexion's tendency to potentially revise initially correct answers into incorrect ones. Furthermore, in contrast to their behavior in the USACO benchmark, retrieval methods, such as standard RAG and CoK, do enhance performance in this task. We attribute this to the shorter con-

Table 2: Results (accuracy) on TheoremQA-Math.

| Method | TheoremQA-Math |
|---|---|
| Standard | 39.81 |
| CoT | 41.75 |
| Reflexion | 40.29 |
| RAG | 44.17 |
| CoK | 45.15 |
| CR-Planner | **53.40** |

text of the retrieved documents in the math domain. With shorter retrieved documents, the base model is easier to determine which information to incorporate. Nevertheless, CR-Planner maximizes the benefits of both retrieval and reasoning, leading to the best performance improvement.

### 3.4 REASONING-HEAVY DOMAIN RETRIEVAL

**StackBio and StackEcon.**  Complex domain queries often demand in-depth reasoning to identify relevant documents that go beyond simple surface-level matching. To evaluate models' ability in reasoning-heavy domain retrieval, we use biology- and economics-related queries from the BRIGHT benchmark (Su et al., 2024), specifically StackBio and StackEcon. Both StackBio and StackEcon contain 103 questions sourced from StackExchange, with the gold labels being the documents cited in the answers. As the evaluation metric is nDCG@10, which requires the top 10 documents, we set the number

Table 3: Results on complex domain retrieval.

| Method | StackBio nDCG@10 | StackEcon nDCG@10 |
|---|---|---|
| BM25 | 19.20 | 14.90 |
| CoT | 21.06 | 16.33 |
| CoK | 20.82 | 17.45 |
| CR-Planner | **29.51** | **22.80** |

of retrieved documents to 10 when PC-Planner performs the final retrieval.

**External knowledge.**  In line with the BRIGHT benchmark (Su et al., 2024), external sources can include any accessible web content such as articles, tutorials, news, blogs, and reports. Since this information has already been gathered and incorporated into the benchmark, we employ BM25 for document retrieval to ensure efficiency.

**Results and observations.** As shown in Table 3, CR-Planner consistently improves over the standard BM25 method by 10.31% and 7.9% on StackBio and StackEcon, respectively. CoK improves the standard BM25 method, which indicates that reasoning before retrieval is crucial in such reasoning-heavy domain retrieval tasks. However, CoK does not consistently enhance performance; for instance, it performs worse than CoT on StackBio. We attribute this to the potential noise introduced by multiple suboptimal retrieval results. These observations further highlight the effectiveness of the critic models in RC-Planner.

## 4 ANALYSIS

### 4.1 DOMAIN-SPECIFIC CRITIC MODELS

The critic models play a key role in CR-Planner by guiding the selection of sub-goals and executions through inference. Previous works often adopt proprietary LLMs as critic models (*e.g.,* GPT-4 and Claude), utilizing in-context learning to evaluate actions (Gou et al., 2024; Zhao et al., 2024). In this subsection, we compare CR-Planner's performance when using either fine-tuned models or GPT-4 (`gpt-4o-2024-05-13`) as critics on the USACO and StackBio datasets. The results are presented in Figure 4.1. Although employing GPT-4 as the critic yields improvements over the baseline, CR-Planner consistently performs better with fine-tuned critic models. Notably, the fine-tuned critic models lead to larger gains in tasks

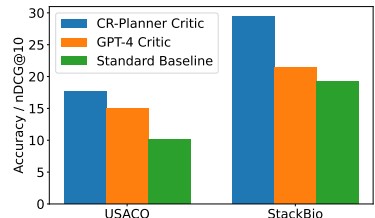

Figure 3: Performances of different critic models vs. baseline.

that require domain knowledge, such as StackBio. This underscores the significance of domain-specific fine-tuning and the rationale behind CR-Planner's use of fine-tuned critic models.

### 4.2 FLEXIBILITY OF CRITIC MODELS ON VARIOUS BASE MODELS

Table 4: CR-Planner with various base models.

| Method | USACO |
|---|---|
| Claude-3.5 | 9.12 |
| CR-Planner w/ Claude-3.5 | 13.68 |
| Llama-3.1 | 7.49 |
| CR-Planner w/ Llama-3.1 | 10.10 |

Compared to previous methods like Self-RAG (Asai et al., 2024), CR-Planner does not require fine-tuning the base model. This flexibility allows CR-Planner to be applied across various base models, whether open-source or closed-source. In this subsection, we showcase the effectiveness of our critic models on another closed-source model, Claude-3.5 (`claude-3-5-sonnet`), and an open-source model, Llama-3.1 (`Llama-3.1-70B-Instruct`). As demonstrated in Table 4, CR-Planner enhances both Claude-3.5 and Llama-3.1. However, the improvements, 4.56% for Claude-3.5 and 2.61% for Llama-3.1, are smaller compared to the 7.49% boost seen with GPT-4. We believe this is due to the critic models being trained on data collected from GPT-4, making them more attuned to GPT-4 during inference and potentially less optimized for other models. Nonetheless, the plug-and-play nature of critic models in our CR-Planner presents a promising approach to distill planning capabilities from powerful LLMs. This planning ability can be utilized to directly guide smaller LLMs, which lack the strength to generate high-quality MCTS trajectories on their own.

### 4.3 RETRIEVE OR NOT TO RETRIEVE

Tackling complex domain-specific tasks such as competitive programming requires extensive reasoning as well as advanced algorithmic knowledge, which base models may not inherently possess. In this section, we examine the importance of accurately retrieving external knowledge to assist in solving competitive programming problems. We instruct the model to concentrate solely on reasoning, employing the reasoning critic model $g_{\text{REASON}}^g$ to select a rationale

Table 5: CR-Planner with and without retrieval.

| Method | USACO |
|---|---|
| Standard | 10.10 |
| CR-Planner w/o Retrieval | 14.33 |
| CR-Planner | 17.59 |

for each reasoning step. As shown in Table 5, the performance without retrieval is lower. However, as discussed in Section 3.2 and by Shi et al. (2024), inaccurate retrieval could impair performance. This emphasizes the critical role of accurate retrieval and the overall effectiveness of CR-Planner.

## 4.4 Execution Sampling

Throughout both the training and inference phases of CR-Planner, executing sub-goals involves sampling several possible candidates. By increasing the number of candidates, the likelihood of identifying a better option may improve. In this subsection, we study how changing the number of candidates sampled for sub-goal execution during inference affects performance. However, due to cost concerns, we do not conduct ablation studies for the training phase. As illustrated in Figure 4.4, the improvements on USACO are substantial when increasing from one to two, but converge around three. We believe this is due to the limitations of the generator model's reasoning capabilities and the retriever's accuracy. Without fine-tuning both generator and retriever models, further performance gains would be challenging to achieve. Therefore, to balance between performance and cost, we select three as the sampling number for the inference of main experiments.

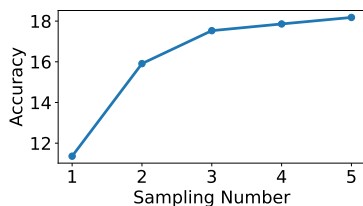

Figure 4: Performances of various execution sampling.

## 5 Related Work

LLMs have demonstrated inherent reasoning capabilities, showing promising performance in most logical reasoning datasets (Liu et al., 2023; Qin et al., 2023). However, using standard LLMs directly often fall short in complex reasoning tasks that require structured thinking or planning (Huang & Chang, 2023). Therefore, researchers have been attempting to develop more sophisticated reasoning schemes. Chain-of-Thought (CoT) (Wei et al., 2022) prompts LLMs to articulate the reasoning processes step by step, improving performances on complex tasks. Tree-of-Thought (Yao et al., 2023a) then generalizes further by breaking down a CoT into coherent units of "thoughts", thus enabling the LLM to consider multiple reasoning paths and self-evaluate to decide the next course of action. To further improve LLMs in planning-based reasoning, research finds that process supervision shows a promising way forward. RAP (Lightman et al., 2024) uses a world model to estimate future rewards of reasoning steps, providing step-wise guidance for reasoning processes. Jiao et al. (2024) learns planning-based reasoning through direct preference optimization (DPO) (Rafailov et al., 2023) on collected trajectories, which are ranked according to synthesized process rewards. As a result, tuned 7B models can surpass GPT-3.5-Turbo. However, this method requires training the base model, which limits its applicability to larger and closed-source models. In comparison, CR-Planner trains external critic models, which offers flexibility for use with any base model.

Besides reasoning improvements, retrieval augmented generation (RAG) can effectively reduce hallucinations (Huang et al., 2023) by introducing external knowledge. Specifically, the RAG process can be divided into 3 sub-tasks: pre-retrieval analysis, query generation and rewriting, and document selection. Currently, most methods attempt to optimize the subtasks seperately. Self-ask (Press et al., 2023) optimizes pre-retrieval analysis by breaking down the original problem into sub-problems. Chain of Knowledge (Li et al., 2024) rewrites natural-language questions to database queries for more precise retrieval with structured knowledge. RePlug (Shi et al., 2023) improves document selection with a fine-tuned retriever. As these methods optimize sub-tasks locally, the single-task improvements may not constitute the globally optimal solution. In comparison, CR-Planner trains the critic model by learning the rewards of each individual action for overall performance.

## 6 Conclusions

In this paper, we present critic-guided planning with retrieval-augmentation (CR-Planner), a novel framework for handling domain-knowledge-specific and reasoning-heavy tasks by leveraging fine-tuned critic models to guide both the reasoning and retrieval processes. We further employ the Monte Carlo Tree Search for systematic data collection to enhance the training of the critic models. Our approach, validated across challenging domains like competitive programming, math reasoning, and complex domain retrieval tasks, has shown substantial performance improvements over existing methods. By combining the strengths of large generalist models with domain-specific fine-tuned critics, CR-Planner offers a flexible and scalable solution for solving problems that require both intricate reasoning and accurate knowledge retrieval.

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

## A   PROMPTS USED IN DIFFERENT METHODS

### A.1   RC-PLANNER (COMPETITIVE PROGRAMMING)

#### A.1.1   INSTRUCTION

Reason through the problem and think step by step. Specifically:
1. Restate the problem in plain English.
2. Conceptualize a solution first in plain English.
3. Write a pseudocode solution
4. Output the Python 3 solution to the problem. Make sure to wrap your code in "'python and "'
Markdown delimiters, and include exactly one block of code with the entire solution.
No outside libraries are allowed.

[BEGIN PROBLEM]

[END PROBLEM]

#### A.1.2   SUBGOAL SELECTION

To proceed, below are the available actions:

[REASON] - Provide a reasoning step.

[GENQUERY] - Generate a query to retrieve information from external knowledge sources.

[RETRIEVE] - Retrieve documents using the query.

The next step is [].

#### A.1.3   EXECUTION SELECTION - RATIONALE SAMPLING

Reason through the problem and think step by step. Specifically:
1. Restate the problem in plain English.
2. Conceptualize a solution first in plain English.
3. Write a pseudocode solution
4. Output the Python 3 solution to the problem. Make sure to wrap your code in "'python and "'
Markdown delimiters, and include exactly one block of code with the entire solution.
No outside libraries are allowed.

[BEGIN PROBLEM]

[END PROBLEM]

Generate one next reasoning step (*e.g.,* [BEGIN REASON] Restate the problem: ... [END REA-SON]). It starts with [BEGIN REASON] and ends with [END REASON]. Do not include the subsequent reasoning steps.

#### A.1.4   EXECUTION SELECTION - QUERY SAMPLING

To verify or solve the reasoning step, I need additional information from external knowledge sources (*e.g.,* textbook). And I need to generate a query to get that information. The query needs to be conceptual but relevant to the reasoning step. The query should not contain any specific numbers or entities of the reasoning step. The query starts with [BEGIN QUERY] and ends with [END QUERY]. Stop the generation when the query is completed.

[BEGIN REASON]

[END REASON]

### A.2   CoT

Reason through the problem and think step by step. Specifically:
1. Restate the problem in plain English.
2. Conceptualize a solution first in plain English.
3. Write a pseudocode solution
4. Output the Python 3 solution to the problem. Make sure to wrap your code in "'python and "'
Markdown delimiters, and include exactly one block of code with the entire solution.
No outside libraries are allowed.

[BEGIN PROBLEM]

[END PROBLEM]

### A.3   CHAIN-OF-KNOWLEDGE

#### A.3.1   REASONING GENERATION

Reason through the problem and think step by step. Specifically:
1. Restate the problem in plain English.
2. Conceptualize a solution first in plain English.
3. Write a pseudocode solution
4. Output the Python 3 solution to the problem. Make sure to wrap your code in "'python and "'
Markdown delimiters, and include exactly one block of code with the entire solution.
No outside libraries are allowed.

[BEGIN PROBLEM]

[END PROBLEM]

#### A.3.2   RATIONALE CORRECTION

The given sentence may have errors, please correct them based on the given external knowledge.

Sentence: [Rationale]
Knowledge: [Knowledge]
Edited sentence:

#### A.3.3   NEXT RATIONALE GENERATION

Reason through the problem and think step by step. Specifically:
1. Restate the problem in plain English.
2. Conceptualize a solution first in plain English.
3. Write a pseudocode solution
4. Output the Python 3 solution to the problem. Make sure to wrap your code in "'python and "'
Markdown delimiters, and include exactly one block of code with the entire solution.
No outside libraries are allowed.

[BEGIN PROBLEM]

[END PROBLEM]

[START PRECEDING RATIONALES]

[END PRECEDING RATIONALES]

## A.4 REFLEXION

### A.4.1 ACTOR

You are a Python writing assistant. You will be given your previous implementation of a function, a series of unit tests results, and your self-reflection on your previous implementation. Apply the necessary changes below by responding only with the improved body of the function. Do not include the signature in your response. The first line of your response should have 4 spaces of indentation so that it fits syntactically with the user provided signature.

Reflexion Actor generations follow the form:

[Instruction]

[Function implementation]

[Unit test feedback]

[Self-reflection]

[Instruction for next function implementation]

### A.4.2 SELF-REFLECTION

You are a Python writing assistant. You will be given your previous implementation of a function, a series of unit tests results, and your self-reflection on your previous implementation. Apply the necessary changes below by responding only with the improved body of the function. Do not include the signature in your response. The first line of your response should have 4 spaces of indentation so that it fits syntactically with the user provided signature.

Reflexion Self-Reflection generations follow the form:

[Instruction]

[Function implementation]

[Unit test feedback]

## B   A RUNNING EXAMPLE

Below is a running example of CR-Planner. Selected action for each step is highlighted in green :

**Problem:** Given a string $s$, find the length of the longest substring without repeating characters in optimal time complexity.

**Instruction:** Reason through the problem and think step by step. Specifically:
1. Restate the problem in plain English.
2. Conceptualize a solution first in plain English.
3. Write a pseudocode solution
4. Output the Python 3 solution to the problem. Make sure to wrap your code in "'python and '"
Markdown delimiters, and include exactly one block of code with the entire solution.
No outside libraries are allowed.

**Step 1 (Sub-Goal Selection):** REASON : The next step is to generate a rationale; GENQUERY: The next step is to generate a query; RETRIEVE: The next step is to retrieve a document.

**Step 2 (Execution Selection):** RATIONALE 1: The optimal time complexity is $O(n^2)$; RATIONALE 2 : The optimal time complexity is $O(n)$; RATIONALE 3: The optimal time complexity is $O(n^3)$.

**Step 3 (Sub-Goal Selection):** REASON: The next step is to generate a rationale; GENQUERY : The next step is to generate a query; RETRIEVE: The next step is to retrieve a document.

**Step 4 (Execution Selection):** QUERY 1: Given a string $s$, find the length of the longest substring without repeating characters in optimal time complexity; QUERY 2: Sliding window technique string problems; QUERY 3 : Max length substring with unique characters with $O(n)$ complexity.

**Step 5 (Sub-Goal Selection):** REASON: The next step is to generate a rationale; GENQUERY: The next step is to generate a query; RETRIEVE : The next step is to retrieve a document.

**Step 6 (Execution Selection):** DOC 1: Given a string s, find the length of the longest substring without repeating characters. Examples: Input: "ABCBC" Output: 3 Explanation: The longest substring without repeating characters is "ABC" Input: "AAA" Output: 1 Explanation: The longest substring without repeating characters is "A" Input: "GEEKSFORGEEKS" Output: 7 Explanation: The longest substrings without repeating characters are "EKSFORG" and "KSFORGE" with lengths of 7; DOC 2: To deal with time complexity problems, it always helps to scale the problem up and think of a massive case. If your string was thousands of characters long, we still only have one start pointer and one end pointer. The key thing is that both the pointers will only ever move forward (along the string). Therefore, the complexity of this is definitely $O(n)$ since they are only moving forward together through the string - so the time this process would take is proportional to the length of the string (the time it takes to get to the end); DOC 3 : The intuition behind the solution is to iteratively find the longest substring without repeating characters by maintaining a sliding window approach. We use two pointers (left and right) to represent the boundaries of the current substring. As we iterate through the string, we update the pointers and adjust the window to accommodate new unique characters and eliminate repeating characters.

**Step 7 (Sub-Goal Selection):** REASON : The next step is to generate a rationale; GENQUERY: The next step is to generate a query; RETRIEVE: The next step is to retrieve a document.

**Step 8 (Execution Selection):** RATIONALE 1: The retrieved document is not sufficient for solving the problem. Therefore, a second-level retrieval is required; RATIONALE 2 : To solve the problem efficiently, use a sliding window technique: 1. Initialize a Window: Start with a window at the beginning of the string, which represents the current substring without repeating characters. 2. Expand the Window: Move through the string one character at a time, adding each character to a data structure (like a set or dictionary) that keeps track of characters in the current window. 3. Check for Repeats: If you encounter a character that is already in the data structure, it means there's a repetition within the current window. 4. Adjust the Window: Move the start of the window forward, removing characters until the repeated character is excluded from the window. This ensures the window contains only unique characters. 5. Update Maximum Length: Keep track of the maximum size of the window throughout the process. This represents the length of the longest substring without repeating characters. 6. Continue Until End of String: Repeat the expand and adjust steps until you have traversed the entire string. This approach ensures that you examine each character at most twice (once when added and once when removed), resulting in optimal linear time complexity; RATIONALE 3: Here is the code:

```python
def length_of_longest_substring(s):
    char_index = {}
    max_length = 0
    start = 0

    for idx, char in enumerate(s):
        if char in char_index:
            start = char_index[char]
        char_index[char] = idx
        current_length = idx - start
        if current_length > max_length:
            max_length = current_length

    return max_length
```

**Step 9 (Sub-Goal Selection):** REASON : The next step is to generate a rationale; GENQUERY: The next step is to generate a query; RETRIEVE: The next step is to retrieve a document.

**Step 10 (Execution Selection):** RATIONALE 1: Here's a step-by-step plan: Initialize Pointers: Start with two pointers, left and right, both at the beginning of the string. These pointers define the current

Table 6: Details of state types and action spaces.

| Current State $s_t$ | Action Space $\mathcal{A}_{s_t}$ | Next State $s_{t+1}$ |
|---|---|---|
| REASON | temperature sampling | RATIONALE |
| GENQUERY | temperature sampling | QUERY |
| RETRIEVE | top-k candidates | DOC |
| RATIONALE | reasoning, querying, retrieving | REASON / GENQUERY / RETRIEVE |
| QUERY | reasoning, querying, retrieving | REASON / GENQUERY / RETRIEVE |
| DOC | reasoning, querying, retrieving | REASON / GENQUERY / RETRIEVE |

window of unique characters. Create a Character Map: Use a hash map (or an array if dealing with a limited character set like ASCII) to keep track of characters and their latest indices in the string. Iterate Through the String: Move the right pointer through the string one character at a time. a. Check for Repeats: For each character s[right], check if it exists in the character map and if its last recorded index is on or after the left pointer. This indicates a repeating character within the current window. b. Adjust the Left Pointer: If a repeat is found, move the left pointer to one position after the last occurrence of the current character. This effectively skips the repeated character and shortens the window from the left side. c. Update the Character Map: Record or update the index of the current character s[right] in the character map with the current right index. d. Calculate the Maximum Length: Compute the length of the current window as current_length = right - left + 1. Update a variable max_length if current_length is greater than the previously recorded maximum. Continue Until End of String: Repeat step 3 until the right pointer reaches the end of the string. Return the Result: After the loop ends, max_length will contain the length of the longest substring without repeating characters; RATIONALE 2 : Here is the code:

```python
def length_of_longest_substring(s):
    char_index = {}
    max_length = 0
    start = 0

    for idx, char in enumerate(s):
        if char in char_index and char_index[char] >= start:
            start = char_index[char] + 1
        char_index[char] = idx
        max_length = max(max_length, idx - start + 1)

    return max_length
```

RATIONALE 3: Here is the code:

```python
def length_of_longest_substring(s):
    char_set = set()
    left = 0
    max_length = 0

    for right in range(len(s)):
        if s[right] in char_set:
            char_set.clear()
            left = right + 1
        char_set.add(s[right])
        max_length = max(max_length, right - left + 1)

    return max_length
```

## C  CR-PLANNER STATE TYPES AND ACTION SPACES

We provide detailed information on state types and action spaces for CR-Planner in Table 6.