# OpenReview forum: "Can We Further Elicit Reasoning in LLMs? Critic-Guided Planning with Retrieval-Augmentation for Solving Challenging Tasks"
_ICLR.cc/2025/Conference — ICLR 2025 Conference Withdrawn Submission_

### Official Review · Reviewer_va9m · 2024-10-29

**Soundness:** 2
**Presentation:** 3
**Contribution:** 3
**Rating:** 3
**Confidence:** 3

**Summary:**

The paper structures response planning for reasoning-intensive tasks as a series of reasoning steps and knowledge retrieval steps. To guide this complex plan formation, reasoning, query generation and retrieval critics are used to score the execution of sub-goals. A separate sub-goal selection critic is used for informing sub-goal selection. These critics are obtained by fine-tuning an open-source reward model on data collected via MCTS. Evaluations are performed on a competitive programming benchmark, a theorem-driven math benchmark and two reasoning-heavy domain retrieval benchmarks. Further analysis and comparisons are performed on the programming benchmark.

**Strengths:**

The use of stepwise critics seem to address an important problem: guiding the generation of long plans and reasoning traces without compounding errors or other degenerative behaviour. This is especially challenging given the inclusion of RAG, which can help simultaneously address knowledge-intensive aspects of tasks.

The method is thoroughly described and overall very well-communicated.

The results show performance gains over relevant benchmarks, although it is unclear whether the comparisons are fair given domain-specific fine-tuning budgets (see 'Weaknesses').

**Weaknesses:**

Nowhere is it stated what tasks are used for training the critic models. I can therefore only assume that training subsets of the benchmarks used for evaluation are used for training the critics. If so, this makes comparison to the current baselines unfair.

The reliance on domain-specific fine-tuning of several critic models is a notable limitation, especially given that the paper does not compare to simply fine-tuning the generating model with the same budget.

There is no ablation of the individual critics. The paper shows the importance of fine-tuned, domain-specific critics, but does not ablate the inclusion of a critic for each of the sub-goal execution types and sub-goal selection.

There are no ablations or experiments showing the impact, in terms of sample efficiency, of MCTS on critic fine-tuning.

**Questions:**

What were the domain-specific tasks used for training the critics?

Given the need for training domain-specific critics, why not simply fine-tune a domain-specific generator? Perhaps the explicit planning, following the structure of CR-Planner gives a bigger performance boost, but empirically verifying this is important.

What was used to motivate the separate fine-tuning of four domain-specific critics? Did you try creating one dataset for all of the critic tasks and fine-tuning a single critic? Results on this would be useful.

---

### Official Review · Reviewer_3JmH · 2024-11-01

**Soundness:** 1
**Presentation:** 1
**Contribution:** 1
**Rating:** 3
**Confidence:** 4

**Summary:**

The paper presents CR-Planner (Critic-guided Planning with Retrieval-Augmentation), a framework designed to enhance reasoning and factual correctness in large language models (LLMs) by employing critic models for sub-goal selection and execution guidance. Traditional methods like chain-of-thought (CoT) reasoning and retrieval-augmented generation (RAG) have proven effective in structured reasoning tasks, but struggle with complex challenges in areas such as competitive programming and theorem-driven mathematics. CR-Planner addresses these challenges by using fine-tuned critic models to guide reasoning and retrieval, enabling a systematic selection and execution of sub-goals. Monte Carlo Tree Search (MCTS) is utilized to train data collection. Experiments are across competitive programming, math problems, and domain retrieval.

**Strengths:**

1. The paper evaluates CR-Planner on diverse, challenging domains, demonstrating substantial improvements in both reasoning-intensive and retrieval-intensive tasks.
2. By utilizing MCTS for data generation, CR-Planner effectively addresses data scarcity in training critic models, which is a significant practical contribution.

**Weaknesses:**

1. A key concern with the proposed approach is the soundness of using a critic model for action selection. While the critic model is trained to optimize the reward function, this function alone does not constitute an actionable policy, potentially limiting the framework's effectiveness. This limitation is further complicated by the mixed action space, which combines both continuous and discrete elements in execution.

2. The paper would benefit from substantial revisions to provide clearer explanations of each component. In particular, detailed descriptions are needed for the problem formulation, the critic model’s training methods, and the use of pairwise ranking loss, along with a comprehensive equation and explanation. Additionally, the data collection process for each critic model requires clarification to enhance transparency and reproducibility.

3. The novelty of the proposed approach is limited, as the use of MCTS for reasoning is well-established in existing literature. The lack of comparisons with related works on MCTS-based reasoning approaches makes it difficult to assess the unique contributions of this paper within the field.

**Questions:**

1. Could the authors clarify the volume and scope of data required for effective critic model training?
   - Is the data collection process designed to be online or offline?
   - Are there distinct types of data required for training each critic model?
   - To what extent is the trained critic model generalizable across different tasks or domains?

2. The paper (lines 149-151) notes that MCTS is used to gather training data that captures global reward information. However, in large language models (LLMs) where the action space is effectively infinite, selecting optimal actions solely based on a value function may be impractical, as an optimal action cannot be deduced simply by comparing all possibilities.
   - How do the authors address the challenges posed by this expansive action space?
   - Given that value functions alone may not guide optimal action selection in LLM settings, what alternative strategies or adaptations are employed?

3. The paper mentions temperature sampling for the REASON and GENQUERY actions and top-k candidates for the RETRIEVE action, yet it is unclear how these discrete and continuous spaces are integrated.
   - Could the authors provide a clearer description of the combined action space used in execution selection, beyond the reference to Appendix C and Table 6?

4. In Equation 1, the policy selects actions based on the reward function rather than the value or action-value function, which may not yield the optimal trajectory.
   - Could the authors explain the rationale for using the reward function over other potential selection criteria that might better optimize the full trajectory?

5. The section "Collecting data via MCTS" outlines MCTS with some modified definitions, yet lacks specifics.
   - How does the approach balance exploration within unknown search spaces?
   - Is data collected in a single batch or in increments, and what is the distribution among different data types collected?

6. The experiment section does not address other relevant work on MCTS in reasoning contexts.
   - Could the authors discuss their approach in relation to similar works, such as:
     - *Q*: Improving Multi-step Reasoning for LLMs with Deliberative Planning. http://arxiv.org/abs/2406.14283
     - Quiet-STaR: Language Models Can Teach Themselves to Think Before Speaking. http://arxiv.org/abs/2403.09629
     - Monte Carlo Tree Search Boosts Reasoning via Iterative Preference Learning. http://arxiv.org/abs/2405.00451

---

### Official Review · Reviewer_nVAp · 2024-11-03

**Soundness:** 2
**Presentation:** 3
**Contribution:** 2
**Rating:** 6
**Confidence:** 3

**Summary:**

The paper proposes a novel planning framework called Critic-guided planning with Retrieval-augmentation (CR-Planner). Due to frequent reasoning errors and irrelevant knowledge retrieval in planning, existing LLM-based methods that leverage techniques like Chain-of-Thought (CoT) and Retrieval-Augmented Generation (RAG) often struggle with reasoning-heavy and domain-knowledge-intensive tasks, such as competitive programming and mathematics. To address this, CR-Planner fine-tunes critic models using data collected from Monte Carlo Tree Search (MCTS) to evaluate and execute answers at each stage of planning, thereby enhancing the reasoning capabilities of planning systems. Experiments conducted on challenging tasks that require domain knowledge and reasoning, including competitive programming, theorem-driven mathematical reasoning, and complex domain retrieval problems, demonstrate that CR-Planner outperforms the baseline by an average of 10.06%.

**Strengths:**

Strength 1: The paper aims to solve challenging reasoning problems, such as those found in programming and theorem-driven mathematical reasoning, which is a meaningful endeavor. Additionally, the experimental performance that exceeds baselines by an average of 10.06% is striking.
Strength 2: The motivation to use fine-tuned critics for evaluating planning answers is intuitively reasonable. Furthermore, identifying the source of planning errors—whether in reasoning or retrieving—and proposing an approach that breaks down the planning process into Reason, GenQuery, and Retrieve is both impressive and rational.
Strength 3: Overall, the article is well-written, with clear articulation and methodological figures that effectively convey the content.

**Weaknesses:**

Weakness 1: The training data used for fine-tuning the Critic is collected via MCTS, which imposes a heavy computational load. Moreover, the nodes generated during MCTS are sampled from LLM. Consequently, the process of labeling data by MCTS and LLM is costly.
Weakness 2: Intuitively, there is a temporal relationship among the three sub-goals: Reason, GenQuery, and Retrieve. The system must first engage in reasoning, then generate a query based on the reasoning result, and finally retrieve the result based on the generated query. Therefore, the idea of fine-tuning a critic for sub-goal selection seems somewhat unnecessary.

**Questions:**

uestion 1: Although decomposing the planning process into three stages—Reason, GenQuery, and Retrieve—seems intuitively reasonable, it appears that there are no experimental results provided to validate this approach. For example, is there a need to fine-tune an additional critic for evaluating and executing the sampling answers for the "Standard" method in baselines? Can the authors provide such results and compare them with those of the CR-Planner?

Question 2: Is there indeed a necessity to fine-tune a critic for Sub-Goal Selection? It seems that there is a certain temporal relationship among the stages of Reason, GenQuery, and Retrieve: the system must first reason, then generate a query based on the reasoning result, and finally retrieve the result based on the generated query. Are there scenarios where this temporal sequence does not apply?

---

### Official Review · Reviewer_3G35 · 2024-11-04

**Soundness:** 3
**Presentation:** 3
**Contribution:** 2
**Rating:** 3
**Confidence:** 4

**Summary:**

This paper introduces CR-Planner, a method combining critic-guided planning with retrieval augmentation to tackle reasoning-heavy tasks. The primary contribution of this work is the use of fine-tuned critic models to assist in action selection during inference, with training data for the critic models derived from offline Monte Carlo Tree Search (MCTS) data collection. Experimental results indicate that the proposed method outperforms baseline approaches.

**Strengths:**

* The paper is clearly written and easy to follow, with a well-articulated description of the proposed approach. The visualizations are also clear and supportive of the narrative.
* The proposed method demonstrates superior performance compared to baseline methods.

**Weaknesses:**

* While the performance improvement over baselines is evident, it is not entirely surprising. The critic essentially distills the MCTS policy from GPT-4 and is expected to perform better than the original policy, given the presence of a reliable external verifier. Moreover, the comparison between CR-Planner and other baselines may not be fair, as CR-Planner likely incurs significantly higher computational costs due to the extensive computations required for MCTS offline data collection.
* The training prompts for the critic model appear to be identical to the test tasks, which raises concerns about the generalizability of the learned critic. It would be beneficial to demonstrate the performance of the learned critic on different datasets to establish its ability to generalize beyond the training data. Otherwise, the model may merely approximate the offline MCTS performance.
* The authors should also include the performance metrics of the vanilla MCTS policy to provide a baseline comparison. This would help quantify the extent of performance degradation when using the learned neural approximator (i.e., the critic).
* To this matter, the technical contributions presented in this work seem to be limited.

Overall Assessment:
While the paper presents a well-written and clear description of CR-Planner and shows promising performance improvements, the evaluation lacks important details regarding computational fairness and generalizability, as well as its technical contribution limitation.  Consequently, I recommend rejection to this submission.

**Questions:**

N/A

---

### Note · Authors · 2024-11-25

**Comment:**

We appreciate the reviewers' feedback and suggestions and will incorporate the relevant ones into our next version.

**Withdrawal Confirmation:**

I have read and agree with the venue's withdrawal policy on behalf of myself and my co-authors.